# Conofolidine: A Natural Plant Alkaloid That Causes Apoptosis and Senescence in Cancer Cells

**DOI:** 10.3390/molecules29112654

**Published:** 2024-06-04

**Authors:** Mohammed Zuhair Al-Hayali, Choy-Eng Nge, Kuan Hon Lim, Hilary M. Collins, Toh-Seok Kam, Tracey D. Bradshaw

**Affiliations:** 1School of Pharmacy, Al-Kitab University, Kirkuk 36015, Iraq; 2School of Pharmacy, Biodiscovery Institute, University of Nottingham, University Park, Nottingham NG7 2RD, UK; hilary.collins@nottingham.ac.uk; 3Department of Chemistry, Faculty of Science, University of Malaya, Kuala Lumpur 50603, Malaysia; choyeng86@gmail.com (C.-E.N.); tskam@um.edu.my (T.-S.K.); 4School of Pharmacy, University of Nottingham Malaysia, Jalan Broga, Semenyih 43500, Malaysia; kuanhon.lim@nottingham.edu.my

**Keywords:** conofolidine, bisindole alkaloids, carcinoma cell lines, cytotoxicity, apoptosis, senescence, ROS generation, γ-H2AX

## Abstract

Natural products contribute substantially to anticancer therapy; the plant kingdom provides an important source of molecules. Conofolidine is a novel *Aspidosperma-Aspidosperma* bisindole alkaloid isolated from the Malayan plant *Tabernaemontana corymbosa*. Herein, we report conofolidine’s broad-spectrum anticancer activity together with that of three other bisindoles—conophylline, leucophyllidine, and bipleiophylline—against human-derived breast, colorectal, pancreatic, and lung carcinoma cell lines. Remarkably, conofolidine was able to induce apoptosis (e.g., in MDA-MB-468 breast) or senescence (e.g., in HT-29 colorectal) in cancer cells. Annexin V-FITC/PI, caspase activation, and PARP cleavage confirmed the former while positive β-gal staining corroborated the latter. Cell cycle perturbations were evident, comprising S-phase depletion, accompanied by downregulated CDK2, and cyclins (A2, D1) with p21 upregulation. Confocal imaging of HCT-116 cells revealed an induction of aberrant mitotic phenotypes-membrane blebbing, DNA-fragmentation with occasional multi-nucleation. DNA integrity assessment in HCT-116, MDA-MB-468, MIAPaCa-2, and HT-29 cells showed increased fluorescent γ-H2AX during the G1 cell cycle phase; γ-H2AX foci were validated in HCT-116 and MDA-MB-468 cells by confocal microscopy. Conofolidine increased oxidative stress, preceding apoptosis- and senescence-induction in most carcinoma cell lines as seen by enhanced ROS levels accompanied by increased NQO1 expression. Collectively, we present conofolidine as a putative potent anticancer agent capable of inducing heterogeneous modes of cancerous cell death in vitro, encouraging further preclinical evaluations of this natural product.

## 1. Introduction

The impact of natural products on new medicines’ discoveries has been indispensable. Over the past 30 years, molecules inspired by nature constituted the mainstay of chemotherapy [1]. Moreover, natural products continue to be discovered, providing novel lead structures which serve as templates for the synthesis of more potent and safer compounds [2,3,4,5], highlighting the importance of continued research for new anticancer agents from nature.

Apoptosis-based chemotherapy constitutes a core focus for anticancer drug development, as achieving this goal is expected to profoundly improve clinical response [6,7]. However, apoptosis evasion is a cancer hallmark [8], and non-apoptotic death mechanisms, e.g., senescence, have also been shown to improve the clinical outcome of anticancer chemotherapy [9]. While apoptosis can induce the early killing of rapidly dividing cancer cells by high doses of anticancer agents, tumour recurrence, drug resistance, and normal tissue damage are common complications [9]. Alternatively, senescence may exert its tumour-suppressive effect at sub-apoptotic doses via the induction of a permanent cell cycle arrest, offering an effective strategy against cancer with fewer adverse effects [6,10]. Indeed, senescence-induction and subsequent senolysis (elimination of senescent cells) is emerging as a convincing strategy in cancer therapy, and plant-derived natural products (such as apigenein) have a role in this regard [11].

Malaysian rainforest species, *Tabernaemontana corymbosa* (Apocynaceae), produces a rich source of promising anticancer agents including jerantinines A and B; jerantinines A and B possess potent antiproliferative activities via the suppression of tubulin polymerization [12,13]. Also isolated from this species are conophylline [14,15], leucophyllidine [16], and bipleiophylline [17] (Figure 1). The aim of the research reported herein was to investigate the antitumour activity of these alkaloids, specifically focusing on novel *Aspidosperma-Aspidosperma* bisindole conofolidine, and additionally interrogating its molecular mechanisms of activity.

We report herein the broad-spectrum in vitro antitumour activity of conofolidine, revealing for the first time its dual apoptosis- and senescence-inducing abilities. Conofolidine’s indole moieties closely resemble those of jerantinines A and B. In this study, we demonstrate conofolidine’s anticancer activity, together with that of conophylline, leucophyllidine, and bipleiophylline (Figure 1) against human-carcinoma cell lines derived from breast, colorectal, lung, and pancreatic carcinomas and B-cell lymphomas. In 2016, Nge et al. reported the isolation and structural elucidation of conofolidine (as a new member of the conophylline group) and the preliminary growth inhibitory effects [18]. In this study, further anticancer activity, cytotoxicity, cell cycle perturbation, and the triggered death pathways (apoptosis and senescence) were investigated after the treatment of carcinoma cells with conofolidine. Preliminary data showed remarkable potency against carcinoma cells, e.g., GI_50_ = 0.054 μM, MCF-7. Effects against foetal lung fibroblasts (MRC-5) and normal breast epithelial cells (MCF10A) were assessed for the preliminary examination of potential selectivity towards carcinoma cells. Mode of action studies led to the identification of reactive oxygen species (ROS) production as one possible mechanism of conofolidine’s growth-inhibitory actions. Furthermore, increased DNA double strand breaks (DSBs) were detected, accompanied by the perturbation of essential cell cycle regulatory protein expression-CDK2, p21, cyclins A2, and D1 in a time-dependent fashion.

## 2. Results

### 2.1. Conofolidine Potently Inhibits Growth of Cancer Cells

The growth inhibitory properties of conofolidine (in addition to conophylline, leucophyllidine, and bipleiophylline) were investigated by MTT assays. All cancer cell lines exhibited significant (*p* ≤ 0.05) sensitivity to conofolidine (Table 1, Figure 2). Conofolidine remarkably suppressed oestrogen receptor-positive (ER+) MCF-7 growth (lowest GI_50_ = 0.054 μM). Triple-negative breast MDA-MB-468, colorectal HT-29, and pancreatic MIAPaCa-2 cells displayed similar GI_50_ values of ~0.3 μM. A549 lung and HCT-116 colorectal carcinoma cells were the least sensitive (GI_50_ = 0.405 and 0.454 μM, respectively). Conophylline, leucophyllidine, and bipleiophylline were less potent, yielding GI_50_ values of 0.07–1.7, 2.0–4.5, and 3.5–13.5 μM, respectively, in the same carcinoma cell lines (Appendix A). The DMSO vehicle did not affect cancer cell growth at the dilutions used (Appendix A).

To confirm conofolidine’s potency and investigate its antitumour spectrum, additional cell lines were included in the MTT assays (Appendix A, Appendix A). Breast carcinoma cell lines were very sensitive to conofolidine (GI_50_ values ≤ 0.17 μM); PANC-1 was similarly sensitive as MIAPaCa-2; the lowest sensitivity was observed in Caco-2 cells (GI_50_ = 0.417 μM). Conofolidine’s antiproliferative effect was also tested against two B-cell lymphoma cell lines (DoHH2 and Vallois), and their GI_50_ values were ≤0.127 μM (Appendix A, Appendix A). To study conofolidine’s selectivity on cancerous compared to noncancerous cells, MRC-5 and MCF10A were used as in vitro models of foetal lung fibroblasts and proliferative benign fibrocystic epithelial breast cells, respectively (Appendix A, Appendix A). Conofolidine’s GI_50_ in these cell lines were ~1.0 μM for MRC-5 and 0.23 μM for MCF10A.

### 2.2. Conofolidine Potently Inhibits Colony Formation

Clonogenic assays were performed to examine whether conofolidine can inhibit cancer cell colony formation. Guided by MTT assay results, two conofolidine concentrations were selected (0.1 and 0.5 μM). Conofolidine potently inhibited cancer cells’ ability to form colonies (Figure 3). Breast MDA-MB-468 and pancreatic MIAPaCa-2 carcinoma colonies were dramatically decreased (99.9%) at 0.1 μM. At the same concentration, conofolidine suppressed MCF-7 (most sensitive cell line) colony formation by 91%, which, intriguingly, was similar to HCT-116 (less sensitive in the MTT assay). However, comparable clonogenic inhibition profiles were seen in both A549 and HT-29 cells: at concentrations of 0.1 and 0.5 μM, conofolidine inhibited colony formation by ~30 and ~95%, respectively. Average plating efficiencies were A549 (68%), HCT-116 (45%), HT-29 (55%), MCF-7 (42%), MDA-MB-468 (28%), and MIAPaCa-2 (51%).

### 2.3. Conofolidine Caused Potent Perturbation of Cell Cycle Progression

After confirming conofolidine’s ability to inhibit cancer cell proliferation and colony formation, we sought to determine whether its mechanism of action involved cell cycle perturbation. Cell cycle analysis was conducted in A549, HCT-116, HT-29, MCF-7, MDA-MB-468, and MIAPaCa-2 after treatment with respective conofolidine 1× and 2× GI_50_ concentrations for 24 and 72 h (Figure 4 and Figure 5). Conofolidine caused significant (*p* ≤ 0.05) cell cycle perturbation, including universal S-phase depletion, especially at 2× GI_50_ treatment, declining to ≤10% in HCT-116, HT-29, MCF-7, and MDA-MB-468 and ≤15% in A549 and MIAPaCa-2 compared to 20–25% for untreated controls in all cell lines. Accompanying S-phase depletion, G1 arrest was observed, either sustained in HT-29 and MCF-7 or temporarily in HCT-116 and MDA-MB-468; A549 exhibited a decreased S-phase without G1 arrest. Interestingly, MIAPaCa-2 cells exhibited G2/M arrest while other cell lines exhibited significantly decreased G2/M events. After a 24 h 1× GI_50_ conofolidine treatment, G1 arrest accompanied S-phase reduction in HCT-116 (~48% from control 40%), HT-29 (73% from control 55%), MCF-7 (~70% from control 46%), and MDA-MB-468 (~65% from control 41%). At 2× GI_50_, 24 h exposure, comparable G1 increments were seen: HCT-116 (50%), HT-29 (73%), MCF-7 (67%), and MDA-MB-468 (62%). In MIAPaCa-2 cells, S-phase depletion was followed by an accumulation of G2/M events after 2× GI_50_, 24 and 72 h exposures: (~23% from control 14%). A less-pronounced arrest in MIAPaCa-2 G2/M phase was seen at 1× GI_50_, 24 h; however, it increased significantly after a 72 h exposure, reaching ~23% from control 14%. Concentration- and time-dependent increased pre-G1 (<2N-DNA) events were seen in all treatment groups of most cell lines except HT-29 and MCF-7 cells, in which significant increments were only observed at 2× GI_50_, 72 h treatment: ~12% from control ~3%. Other cell lines exhibited significant pre-G1 at both 1× and 2× GI_50_ treatments. HCT-116 exhibited the most significant increments, yielding ~20% at 1× GI_50_, 24 and 72 h; and ~30% at 2× GI_50_, 24 and 72 h exposures. In MDA-MB-468 and A549, comparable increases were observed after 72 h, achieving ~17% and ~22% with 1× and 2× GI_50_ treatments, respectively. Intriguingly, MIAPaCa-2 cells showed the lowest significant (*p* ≤ 0.05) pre-G1 events: 8% after 1× GI_50_, 72 h, and ~13% after 2× GI_50_, 24 and 72 h exposures.

### 2.4. Conofolidine-Induced Apoptosis

Considering conofolidine’s ability to suppress proliferation, evoke cytotoxicity in carcinoma cells, and generate pre-G1 populations, we hypothesised that conofolidine may induce cell death via apoptosis. The percentage of apoptotic cells was measured post-conofolidine treatment by flow cytometry following annexin-V-FITC/PI staining. Figure 6 shows that conofolidine induced significant (*p* ≤ 0.05) apoptosis when compared to the controls in three out of six cancer cells: MDA-MB-468, MIAPaCa-2, and HCT-116. These cells displayed significant increases in both early apoptosis (AV+/PI−) and late apoptosis (AV+/PI+). Moreover, both MDA-MB-468 and MIAPaCa-2 exhibited the highest percentages of apoptotic events (AV+) following a 72 h treatment with 2× GI_50_, 54 ± 5.5 and 69 ± 1.1%, respectively, while HCT-116 showed 30 ± 4.7% apoptosis. The percentage of late apoptotic cells (AV+/PI+) was comparable in these cell lines; however, MDA-MB-468 and MIAPaCa-2 displayed massive early apoptosis (AV+/PI−) compared to HCT-116. Other cell lines (A549, HT-29, and MCF-7) did not show significant apoptotic events after treatment with conofolidine 2× GI_50_ values (*p* > 0.05).

### 2.5. Conofolidine-Induced Executioner Caspases and PARP Cleavage

To confirm annexin V-FITC/PI results, we first measured caspases 3/7 activities after 24 h, 2× GI_50_ treatment. As seen in Figure 6c, conofolidine significantly (*p* ≤ 0.05) increased caspases 3/7 activities by ≥2-fold in HCT-116, MDA-MB-468, and MIAPaCa-2 compared to the control. No significant increase in caspases’ activities was observed in A549, HT-29, and MCF-7. Thereafter, we carried out a series of Western blots of whole and cleaved PARP in these cells following 24 and 72 h exposures to 2× GI_50_. HCT-116 displayed cleaved PARP (89 kDa) after a 24 h exposure to conofolidine while less PARP cleavage can be seen in MDA-MB-468 and MIAPaCa-2. After 72 h, these three cell lines showed extensive PARP cleavage. In contrast, A549 and HT-29 display faint cleaved PARP bands only after 72 h while no PARP cleavage can be detected in the caspase 3-deficient MCF-7 cells [19] (Figure 7b).

### 2.6. Conofolidine Promoted Morphological Changes Associated with Senescence Induction

From our observations, conofolidine consistently induced distinctive changes in HT-29 cell morphology and size relative to the control. These changes included cell boundary conversion from rounded and compact to polarised with an increased size and spread relative to the control (Figure 8). The evolution of such phenotype was time-dependent, emerging after 24 h treatment with conofolidine at 2× GI_50_ and continuing throughout the 72 h exposure. After this point, cells started to detach from the supporting matrix, suggesting cell death. When morphological changes accompany S-phase depletion, this could indicate the induction of either differentiation or senescence [20,21]. Accordingly, two hypotheses—differentiation or senescence induction—were postulated. Differentiation was first tested (Appendix A) using sodium butyrate (a known differentiation inducer) and alkaline phosphatase activity (as a differentiation marker). However, the differentiation hypothesis was rejected since conofolidine failed to significantly increase HT-29 alkaline phosphatase activity as the positive control sodium butyrate did. The senescence hypothesis was then considered; thus, β-Gal-histochemical staining was conducted in HT-29 populations in addition to MCF-7 and A549. Following a 72 h exposure to conofolidine 2× GI_50_, these three cell lines developed blue β-Gal staining relative to the control, indicating senescence-induction (Figure 8B). Statistical analyses showed significantly increased senescence-associated β-Gal positive cells (*p* ≤ 0.05) by 50–75% compared to 10–20% for untreated controls in all three cell lines.

### 2.7. Conofolidine Downregulated CDK2, Cyclins (A2, D1), and Upregulated p21

To confirm the cell cycle results, we investigated associated changes in protein expression by Western blotting in selected cell lines (HT-29, HCT-116, and MDA-MB-468) depending on the death pathway triggered: senescence in HT-29 and apoptosis at different time points in HCT-116 and MDA-MB-468. The modulation of the expression of CDK2, cyclin A2, cyclin D1, and p21 was investigated following 24 and 72 h 2× GI_50_ conofolidine treatments (Figure 9). Consistent with cell cycle results, conofolidine induced time-dependent decreases in CDK2 and cyclin A2 expressions relative to the control; CDK2 is required for G1/S-transition, and cyclin A2, upon binding with CDK2, is essential for orderly S-phase progression. Following 24 h, CDK2 expression decreased in these cell lines to ≤60% of the control level. Further reductions were seen after a 72 h treatment, reaching ~20% in HT-29 and MDA-MB-468 and 40% in HCT-116 of the control levels. The decreases in cyclin A2 were more pronounced, reaching ≤30% and ≤10% of the control levels after 24 and 72 h exposures, respectively, in these cell lines. In addition to CDK2 and cyclin A2, G1 progression is controlled by cyclin D1, which activates CDK4/6 upon cell cycle initiation. Conofolidine markedly decreased cyclin D1 in HT-29 to 16% (compared to the control levels) and MDA-MB-468 to 54% after 24 h and to ≤ 33% in both of these cell lines following 72 h; these results corroborated accumulated events in the G1 phase. In contrast, in HCT-116 lysates, cyclin D1 expression was not significantly changed after a 24 h exposure of cells to conofolidine, but it increased to >200% (2-fold) after 72 h compared to the control. The p21 is a known regulator of CDK/cyclin complexes’ activities. Conofolidine significantly increased p21 expression compared to the control in these cell lines. MDA-MB-468 exhibited remarkably increased p21 levels (>100-fold) after 24 and 72 h exposures. Less enhanced p21 expression was seen in both HT-29 and HCT-116: ~3-fold increases following 24 h in both cell lines; 2-fold and 4-fold increments in HT-29 and HCT-116, respectively, after 72 h.

### 2.8. Confocal Images Confirmed the Induction of Aberrant Mitoses by Conofolidine

After confirming conofolidine’s effects on cell cycle regulatory proteins, we validated mitotic abnormities in HCT-116 cells induced by conofolidine by confocal microscopy. The acquisition of images focused on the effects of conofolidine exposures (72 h; 1× and 2× GI_50_) on DNA, tubulin arrangement, and mitoses compared with the control and vincristine (72 h; 10 nM)-treated cells (Figure 10). While untreated cells (a–d) displayed the characteristics of cellular mitosis and cytokinesis, conofolidine-treated ones exhibited aberrant mitotic phenotypes: mitotic slippage, chaotic mitosis (e–g), multipolar spindles (g,h), abnormal spindle, microtubules’ disarrangement (i), membrane blebbing (j), DNA fragmentation (k), and unequal DNA content (m). Compared to vincristine-treated cells (n–p), both induced multipolar spindles and microtubules’ disarrangement; however, vincristine evidently decreased microtubules’ density to a greater extent than conofolidine with abundant multi-nucleation.

### 2.9. Conofolidine Increased the % of γ-H2AX (+) Events

Since DNA fragmentation and p21 upregulation were observed following the exposure of cells to conofolidine, we used flow cytometry to examine whether conofolidine (72 h; 2× GI_50_) is able to induce DNA-DSBs in HCT-116, HT-29, MDA-MB-468, and MIAPaCa-2 cells. Our results showed that conofolidine significant (*p* ≤ 0.05) induced DNA-DSBs following a 72 h exposure in these cell lines (Figure 11 and Appendix A). In HCT-116, 21% of DSBs were observed with conofolidine-treated cells compared to 4% and 49% for the control and etoposide (72 h; 2.0 μM)-treated cells, respectively. MDA-MB-468 cells showed 19% DSBs after conofolidine exposure compared to 3% and 34% for the control and etoposide-treated cells, respectively. In MIAPaCa-2 cells, 15% DSBs were seen in cells exposed to conofolidine (72 h; 2× GI_50_) compared to 3% for the control and 35% for etoposide (72 h; 2.0 μM)-treated cells. However, <10% of DSBs were observed in HT-29 cells compared to 4% for the control and 39% for etoposide-treated cells, respectively. Concurrent cell cycle analyses were performed to determine the cell cycle phase in which the DSBs emerged; interestingly, most DNA-DSBs arose during the G1 phase; MIAPaCa-2 cells revealed some DSBs in the G2/M phase as well.

### 2.10. Conofolidine-Induced γ-H2AX Foci Formation

To confirm the DNA-DSBs flow cytometry results, confocal microscopy was performed in HCT-116 and MDA-MB-468 cells (Figure 11 and Appendix A). Compared to untreated control cells, conofolidine-treated cells (72 h; 2× GI_50_) consistently displayed green fluorescent γ-H2AX foci within the HCT-116 and MDA-MB-468 nuclei. At least one DSBs’ foci saturation per image (i.e., an uncountable number of detectable foci per nucleus leading to foci overlapping) was observed, especially in HCT-116 nuclei. Etoposide (72 h; 2.0 μM)-treated cells displayed increased formation of γ-H2AX foci within their nuclei. Reduced DNA content was seen in both conofolidine- and etoposide-treated cells compared to the control.

### 2.11. Conofolidine-Induced ROS Generation and NQO1 Expression

ROS are chemically reactive molecules that can be generated as redox cycling by-products of quinone derivatives of oxygen-rich anticancer agents, e.g., doxorubicin [22]. The conophylline family members—possibly including conofolidine—are converted to their more active quinone derivatives [23]; hence, they could similarly induce ROS generation. Elevated ROS levels can induce and maintain senescence in cancer cells through a mechanism involving a sustained cell cycle arrest [24]. We hypothesised that conofolidine increased ROS production, contributing to both apoptosis and senescence induction. To test this hypothesis, we measured ROS levels (following 24 h, 2× GI_50_ exposure) in A549, HCT-116, HT-29, MCF-7, MDA-MB-468, and MIAPaCa-2 (Figure 12). ROS generation was modest but significant in colon, lung, and pancreatic cancer cell lines. The most significant (*p* ≤ 0.05) increase was observed in A549 (2-fold), HT-29, and HCT-116 (1.5-fold) relative to the control. A small but significantly elevated ROS level was also observed in MIAPaCa-2 (1.3-fold) and MCF-7 (1.2-fold) relative to the control while untreated and treated MDA-MB-468 cells were ROS-negative. Elevated ROS could induce cellular defence enzymes NQO1, which shows marked inducibility during oxidative stress [25]. Therefore, its expression was validated by Western blot following conofolidine exposure. As expected, NQO1 expression was increased after 24 h in ROS-positive cells: HT-29, HCT-116, A549, MCF-7, and MIAPaCa-2. Similarly, after 72 h, NQO1 expression continued to increase, suggesting that conofolidine efficiently maintained prolonged oxidative stress in these cells. Intriguingly, neither constitutive nor induced NQO1 expression was seen in the MDA-MB-468 cells.

## 3. Discussion

Conofolidine, a novel bisindole alkaloid from *Tabernaemontana corymbosa*, proved its broad-spectrum potency against carcinoma cells derived from breast (MCF-7, MDA-MB-468, MDA-MB-231, SK-BR3, T-47D, ZR-75-B), colon (HCT-116, HT-29, Caco-2), lung (A549), pancreas (MIAPaCa-2, PANC-1), and B-cell lymphoma (DoHH2, Vallois) cell lines (Table 1, Appendix A). The breast carcinoma cells were predominantly the most sensitive to conofolidine. Compared to MRC-5, all carcinoma cell lines exhibited higher sensitivity GI_50_ ≤ 0.5 μM). However, MCF10A—a highly proliferative but benign fibrocystic epithelial breast—cell line showed sensitivity to conofolidine (GI_50_~0.23 μM). MCF-7, MDA-MB-231, T-47D, ZR-75-B, DoHH2, and Vallois cells yielded GI_50_ values ≤ 0.13 μM. Indeed, the potency of conofolidine surpassed structurally-related congener conophylline (Table 1) against A549, MDA-MB-468, and MIAPaCa-2 cell lines. This could be a consequence of the structural variation: the second indole moiety viz. deoxoapodine in conofolidine instead of pachysiphine in conophylline (Figure 1). In addition to its antiproliferative actions, conofolidine efficiently suppressed the clonogenic potential of cancer cells (Figure 3). For example, MDA-MB-468, MIAPaCa-2 colony formation was significantly inhibited >95% by 1 μM conofolidine. Individual cells were unable to survive a 24 h conofolidine challenge and form progeny colonies. A dose-dependent inhibition of A549 and HT-29 colony-forming ability was seen. Together, MTT and clonogenic assays demonstrated conofolidine’s potential to compromise cancer cell proliferation and viability.

To explore conofolidine’s cellular mechanisms of action, cell cycle analysis (using flow cytometry) was considered (Figure 4 and Figure 5). Four distinctive perturbations (specifically at 2× GI_50_) were observed: stark S-phase depletion, G1-phase arrest, G2/M arrest (seen only in MIAPaCa-2), and the emergence of sub-diploid population (<2N-DNA) appearing as pre-G1 events. Following 24 h, conofolidine induced radical S-phase depletion. For example, ≤6% HCT-116, HT-29, MCF-7, and MDA-MB-468 cell populations resided within the S-phase using 1×GI_50_ compared to 20–27% for untreated controls. Moreover, a concomitant increase in G1-populations started to emerge. After 72 h, conofolidine sustained statically significant G1/S-arrest, especially in HT-29 and MCF-7 cells. In contrast, A549 (less sensitive as indicated in previous assays) seemed to require a higher concentration of 2× GI_50_ to produce profoundly equivalent depleted S-phase but without an obvious effect on G1 events. G1-arrest could be a consequence of inhibition of proteins’ transcription and expression, which take place during G1 and are required for DNA replication, ultimately leading to decreased numbers of cells escaping G1-block into S-phase. Similarly, structurally related conophylline was reported to induce G1-arrest in endometrial cancer cells [26]. Thus, structurally related conofolidine and conophylline may share similar anticancer mechanism(s) of action. However, conofolidine was able to decrease the A549 S-phase population without affecting the % of G1-cells, suggesting that it may interfere directly with DNA replication machinery such as replicative helicase similar to anthracyclines [27]. In support of this suggestion, S-phase depletion was attained with minimal effect on G1 in MIAPaCa-2 cells, which interestingly accompanied a G2/M arrest. These observations indicate putative anti-microtubule activity and resemble the effects of microtubule-disrupting agents such as vinca bisindole alkaloids [28] and jerantinines A and B, which share indole moiety structural similarities and are also isolated from *Tabernaemontana corymbosa* [12,13]. Concentration- and time-dependent increased pre-G1 events were evident in most cell lines indicating cell death by apoptosis. For instance, ≥20% of HCT-116 cells accumulated at pre-G1 following 24 h exposure to 1×GI_50_, which continued to increase to ≥30% after 72 h. The depleted S-phase (with changed HT-29 morphology) led to a hypothesis that conofolidine might induce differentiation. Indeed, conophylline is known to differentiate pancreatic-β endocrine cells [29,30]. However, conofolidine, in contrast to the positive control (sodium butyrate), failed to impact HT-29 differentiation (Appendix A).

Considering the sub-diploid events seen in the cell cycle, an apoptotic mode of cell death was the first suggested death pathway to be investigated (Figure 6 and Figure 7). Annexin V-FITC/PI fluorescent staining and flow cytometric analyses of populations were undertaken, which distinguish between early and late apoptosis. In early apoptosis, cell membrane asymmetry is lost, exposing phosphatidylserine to the outer membrane surface. Thus, annexin-V can bind and be measured by flow cytometry. PI can intercalate DNA and be quantified only upon the commencement of late apoptosis when membrane integrity deteriorates [31]. Significantly increased apoptotic events were only observed in MDA-MB-468, MIAPaCa-2, and HCT-116 populations. For instance, >50% of MIAPaCa-2 and MDA-MB-468 were AV+; AV+ events evolve in early apoptosis and are accompanied by PI+ events in late apoptosis when DNA becomes fragmented [32]. Indeed, pre-G1 populations are an indication of DNA cleavage into oligonucleosomal fragments [33]. DNA fragmentation is induced by caspase-activated DNase [7]; thus, caspases activities and the emergence of caspase-mediated cleaved PARP were validated. Indeed, MDA-MB-468, MIAPaCa-2, and HCT-116 showed increased active caspases 3 and 7, which generated the 89 kDa-PARP fragment, corroborating apoptosis. The timing of apoptosis-onset was different between these cells according to the annexin-V/PI results, which were validated by the increased caspase activity and appearance of cleaved PARP at different time points. Intriguingly, A549 cells, which showed significant pre-G1 events, did not display significant AV+ events. These results are consistent with other studies that describe (i) the known apoptosis-reluctant phenotype of A549 cells [34] and (ii) the potential viability of pre-G1 A549 cells [35]. Indeed, conofolidine did not arrest the G1 but decreased S- and G2/M phases, thereby delaying mitosis. Furthermore, these cells might be able to divide, thereby producing two or three cells (tripolar mitosis) possessing sub-diploid DNA content—observed as pre-G1 events. Thus, conofolidine might cause postmitotic arrest in the second and even third cell cycles. Unsurprisingly, the caspase 3-deficient MCF-7 cells did not show any significant AV+ events. This is likely because these cells, as previously reported, have a base pair deletion of caspase 3 gene, which ultimately abrogates mRNA translation [36]. Hence, no apparent cleaved PARP was observed in MCF-7 treated with conofolidine.

Inspired by conofolidine’s potency, G1/S arrest, and altered HT-29 cell morphology (Figure 8) without evident apoptosis or differentiation, we proposed senescence as an alternative pathway. Indeed, increased β-Gal positive HT-29, MCF-7, and A549 cells post-conofolidine exposure supported this proposal (Figure 8B). For example, ≥60% of HT-29 and MCF-7 cells were senescent compared to 15–20% of untreated cells; morphologically, these senescent cells appeared enlarged, flattened, and prolonged. Senescent cells cease to divide—consistent with depleted S-phase—yet they remain metabolically active, staining positively for β-galactosidase, a lysosomal protein and putative senescence biomarker [6]. Cellular senescence may be, in principle, considered a cancer-preventive mechanism, especially in somatic cells that have undergone many divisions and accumulated multiple genetic mutations. By conversion to an immunogenic senescent phenotype, the cell can be eliminated via the immune system [37]. Senescence may also be considered a desired therapeutic outcome in cancer cells able to evade apoptosis through, for example, p53 loss/mutation, occurring in >50% human cancers [38] and exemplified by HT-29 cells. Controversially, persistent senescence may contribute to tumourigenesis [39] through the acquisition of a senescence-associated secretory phenotype (SASP) that may promote tumour progression. However, corroborative evidence demonstrated herein reveals that the ultimate fate of conofolidine-induced senescence is death.

Interference with crucial cell cycle regulators—CDK2, cyclins A2, D1, and p21—is expected with G1/S arrest inducers. Conofolidine affected the expression of these controllers in HT-29, HCT-116, and MDA-MB-468 cells (Figure 9). Downregulation of CDK2 and both cyclin A2 and D1 was observed; as a consequence, these cells were not able to progress G1 through the S-phase. It may be postulated that there was (i) less cyclin D1 available for CDK4/6 activation to phosphorylate the retinoblastoma protein (pRb), thereby triggering G1 initiation [40]; (ii) decreased CDK2 expression required for both G1/S transition—upon binding to cyclin E2—via phosphorylating both histone H1 and pRb [41], and S-phase completion—upon binding to cyclin A2—via phosphorylating the replication machinery components, e.g., DNA polymerase [42]. Diminished cyclin A2, which can activate CDK1 in addition to CDK2, leads to enhanced deterioration of S-phase progression through the perturbation of the DNA replication origin [43], and decreases G2/M transition [44] via disturbing nuclear envelope breakdown, nuclear accumulation of cyclin B1 [45], and anchoring of the mitotic spindle [46]. Thus, the observed G2 arrest accompanying S-phase depletion in MIAPaCa-2 cells may be explained. Our results demonstrating cyclin D1 downregulation in HT-29 and MDA-MB-468 are similar to previously described cyclin D1 suppression by conophylline in carcinoma cells [26]. Conversely, cyclin D1 in HCT-116 was exceptionally increased, possibly because cyclin D1 is available in two isoforms (a and b); cyclin D1b upregulation induces G1 arrest with apoptosis in carcinoma cells [47]. Alternatively, HCT-116 cells might replicate their DNA without completing mitosis and/or cytokinesis and re-enter G1 with tetraploid 4N DNA—driven by the increased cyclin D1—i.e., mitotic slippage. Indeed, confocal microscopy visually confirmed such aberrant division seen as occasional multi-nucleated cells (Figure 10). Upregulated p21 inhibits the kinase activity of CDK2 and CDK2/cyclin complexes [48,49]. In addition, p21 can suppress the proliferating cell nuclear antigen (PCNA) required for DNA replication, thereby blocking S-phase progression [50]. Upregulation of p21 accompanied senescence induction in HT-29 cells. Indeed, p21 mediates cellular senescence via p53-dependent and -independent pathways [51]. However, one could speculate whether p21 overexpression could contribute to apoptosis especially evident in MDA-MB-468. p21 may disrupt transcription factors, e.g., STAT3 [52], which could inhibit the expression of antiapoptotic proteins as some authors reported [53]. Interestingly, increased p21 expression following conofolidine treatment seems to be p53-independent; this is because (i) MDA-MB-468 cells, which displayed a >100-fold p21 increase, harbour a gain of function (GOF)-R273H-mutation in their p53 gene; (ii) HCT-116 cells, which express WT-p53, exhibited only ~3-fold p21 increase. However, recent studies have shown that p21 can be induced by ROS-mediated DNA-damage even when p53 function is compromised [54].

Confocal imaging of HCT-116 cells treated with conofolidine showed several aberrant mitotic phenotypes, supporting our findings (Figure 10). Multinucleation and mitotic slippage (consistent with cyclin D1 upregulation) decreased DNA content, caused microtubules’ disarrangement, and changed polarity (consistent with cyclin A2 and CDK2 downregulation). Furthermore, DNA fragmentation and plasma membrane disintegration and/or blebbing confirmed apoptotic cells (as indicated in annexin V/FITC and cell cycle analyses). The multinucleated cells either succumbed directly to apoptosis or continued another round of division. In support of the second fate, some cells exhibited aberrant, chaotic asymmetric multiplication with spindle disarrangement. Another form of mitotic abnormality appeared as multipolar giant cells with enlarged nuclei, which might be due to the failure of nuclear envelope breakdown. The accumulation of DNA fragments might trigger genotoxic stress that leads to cell death. Similar to vincristine—a microtubule-targeting agent—conofolidine disrupted microtubule and mitotic spindle machinery. However, vincristine decreased microtubule density to a greater extent. DNA fragmentation is a likely consequence of lethal DNA-damage. Indeed, conofolidine caused DNA-DSBs as seen by intense γ-H2AX foci and increased percentage of γ-H2AX+ events, with the majority of events emerging in G1- and G2/M-phases of the cell cycle (Figure 11 and Appendix A). Phosphorylated histone protein H2AX (γ-H2AX) is a robust reporter of unrepaired DNA-DSBs at nascent DSB sites. Accumulation of γ-H2AX in the chromatin around DSB sites creates foci where DNA repair and chromatin remodelling proteins are recruited [55]. Confocal images revealed an increased number of DNA-DSBs foci, with foci saturation frequently seen in HCT-116 cells. In MDA-MB-468 cells, the DNA-damage detected likely precedes exacerbated conofolidine-induced apoptosis. Interestingly, these cells lack a functional pRB, resulting in an attenuated response to DNA damage [56]. Furthermore, conofolidine, by downregulating CDK2 and cyclin A2, halts the functions of both CDK2/cyclin A2 and CDK2/cyclin A1 complexes. Considering the role of A-type cyclins in activating DNA-DSB repair mechanisms in collaboration with CDK2 and Ku DNA repair proteins [57], conofolidine rendered carcinoma cells more susceptible to DNA-damage, leading to aggravated cytotoxicity and apoptosis. Compared to etoposide, a topoisomerase II inhibitor, conofolidine induced significant DSBs at lower concentrations, especially in MDA-MB-468 and MIAPaCa-2 cells: ~0.6 μM conofolidine induced ~20% DSBs compared to ~35% for 2.0 μM etoposide. Thus, conofolidine demonstrated its potency as a DNA-damaging agent comparable to etoposide.

We showed that conofolidine was capable of inducing ROS (H_2_O_2_) in our cancer cell line panel—with the exception of MDA-MB-468 (Figure 12). To cope with oxidative stress, cells increased NQO1 expression, an inducible quinone-detoxifying enzyme. In this regard, all ROS-positive cells exhibited NQO1 upregulation. It is noticeable that NQO1 expression is increased in these cells similarly, although there is variation in their induced H_2_O_2_ level. One could speculate that conofolidine might induce other species of ROS, e.g., hydroxyl radicals (OH^•^), which may trigger NQO1 expression. Oxidative stress, by causing unrepairable DNA-damage, led to p21 upregulation and cell cycle arrest. Thus, ROS may act as mutagens and contribute to the induction and maintenance of cell senescence [58]. Interestingly, some studies reported that NQO1 stabilises p53 or causes p53 accumulation independent of MDM2, provoking senescence and apoptosis [59,60]. ROS are known triggers of apoptosis for plant-derived anticancer agents, e.g., vinca alkaloids [61]. In the case of conofolidine, oxidative stress may elicit apoptosis in HCT-116 and MIAPaCa-2 cells by (i) altering mitochondrial membrane permeability, cytochrome C release, caspase 3/7 activation, and PARP-cleavage; (ii) inducing DNA-damage that is aggravated by conofolidine’s blockade of its repair mechanism via CDK2/cyclin A complex inhibition. The absence of NQO1 in MDA-MB-468 cells is consistent with other studies, which failed to detect NQO1 expression in these cells [62,63]. Such absence might enhance conofolidine’s anticancer action: conofolidine (like other conophylline family members) is converted to its quinone metabolite [18]; accordingly, the absence of the quinone-detoxifying NQO1 delays its metabolic inactivation, enhancing its anticancer activity.

## 4. Materials and Methods

### 4.1. Isolation and Characterization of Conofolidine

All 4 bisindoles were kindly provided by Prof. Kam, University of Malaya. Conofolidine was isolated from the EtOH extract of the leaves of *Tabernaemontana corymbosa* Roxb. ex Wall (Apocynaceae). Its two indoles correspond to taberhanine and 11-hydroxydeoxoapodine, which are connected via a central dihydrofuran ring [18]. The structure and absolute configuration of conofolidine (Figure 1, C_44_H_50_N_4_O_10_, MW = 794.89) were established by spectroscopic methods and X-ray diffraction analysis [18].

### 4.2. Cell Culture

Cancer cell lines were obtained from the American Type Culture Collection (ATCC, Manassas, VA, USA) and were grown and maintained in RPMI-1640 medium supplemented with 10% foetal bovine serum (FBS). Cell lines included were breast carcinoma—MCF-7, MDA-MB-231, MDA-MB-468, SK-BR3, T-47D, and ZR-75-B; colorectal carcinoma—Caco-2, HCT-116, and HT-29; lung carcinoma—A549; pancreatic carcinoma—MIAPaCa-2 and PANC-1; lymphoma—DoHH2 and Vallois. Cells were subcultured twice weekly to sustain their logarithmic growth and were allowed to grow in a humidified atmosphere at 37 °C, 5% CO_2_ incubator. After ≤30 passages, and to reduce phenotypic or genotypic drift, new low-passage cells were reintroduced from cell stocks that had been cryopreserved at −180 °C. Test compound top stocks were kept frozen at −80 °C as 10 mM/10 μL-aliquots in DMSO, which were thawed promptly before use.

### 4.3. MTT Assay

3-(4,5-Dimethylthiazol-2-yl)-2,5-diphenyltetrazolium bromide (MTT, Alfa-Aesar, Lancashire, UK) colourimetric assays were used for testing the antiproliferative ability of test agents on the cancer cells. Cells were seeded into 96-well plates (3 × 10^3^ cells/well) and allowed to attach overnight prior to test compound introduction (5 nM–10 μM, n ≥ 4). Serial dilutions were freshly prepared in the culture medium before the assay. Two measurements of viable cells (at the time of treatment introduction and after a 72 h exposure) were conducted following MTT addition (5 nM–10 μM, n ≥ 4). Plates were then incubated for 3 h, allowing for metabolic reduction of yellow MTT into insoluble purple formazan crystals, which were solubilised in 150 μL DMSO after supernatants’ aspiration. Absorbance readings were recorded at 570 nm in a plate reader (PerkinElmer Life and Analytical Sciences, Buckinghamshire, UK). Dose-response profiles were constructed from mean absorbance values, and test agent concentrations that inhibited growth by 50% (GI_50_) were calculated.

### 4.4. Clonogenic Assay

Clonogenic assays were used to assess the ability of single cells to survive, proliferate, and form colonies following brief exposure to conofolidine. Cells (~400/well) were seeded into 6-well plates in 2 mL medium. After 24 h incubation, conofolidine was introduced (0.1 and 0.5 μM) while the medium alone was added to controls. Following 24 h treatment, medium containing conofolidine was gently aspirated, cells were washed (2× sterile PBS, 37 °C) before fresh conofolidine-free medium was introduced, and plates were re-incubated for 7–9 days. Cells were examined daily until cells in controls had formed colonies consisting of >50 cells. Wells were washed with PBS, fixed with methanol (100%; 15 min), stained with methylene blue (0.5%; 15 min), washed (dH_2_O), and air-dried, and stained colonies were then counted.

### 4.5. Cell Cycle Analysis

The method used was adapted from that described by Nicoletti et al. [64]. Cells were seeded and allowed to attach overnight in 6-well plates (1.0 × 10^5^ and 0.5 × 10^5^ cells/well for 24 h and 72 h exposures, respectively). Following treatment, floating cells were collected, attached cells trypsinised, and cells pooled together in FACS tubes. Samples were washed (PBS; 4 °C) and pelleted by centrifugation (1200 rpm; 5 min; 4 °C). Cells were resuspended in fluorochrome solution (50 μg/mL PI, 100 μg/mL ribonuclease-A, 0.1% triton-X-100, 0.1% sodium citrate). Cell cycle analyses were performed using an FC-500 Beckham-Coulter flow cytometer (Wycombe, UK). Data were collected and analysed using EXPO32 software.

### 4.6. Annexin-V/PI Apoptosis Assay

Cells were seeded at densities of 0.5–0.75 × 10^5^ per well in a 6-well plate, incubated overnight, and treated the following day. After 72 h 2× GI_50_ treatments, cells were harvested, washed (PBS; 4 °C), and transferred with the medium containing detached cells into FACS tubes. After centrifugation (1200 rpm; 5 min; 4 °C) and aspiration of supernatants, cells were resuspended in 5 μL annexin-V-FITC plus 100 μL 1× annexin-V binding buffer (BD Pharmingen). Samples were briefly vortexed and incubated (protected from light for 15 min, 25 °C). PI (10 μL) plus 400 μL annexin-V binding buffer were added to each FACS tube, followed by incubation for 10 min in the dark at 4 °C. Samples were analysed using Beckman Coulter FC-500 flow cytometer and EXPO32 software, and percentages of early apoptotic (annexin-V positive (AV+), PI negative (PI−)) and late apoptotic cells (AV+, PI+) were recorded.

### 4.7. Caspase 3/7 Assay

To measure the executioner caspases’ activity, the Caspase-Glo 3/7 (100 μL; Promega, Madison, WI, USA) assay was conducted according to the manufacturer’s protocol. Cells were seeded at a density of 5 × 10^3^ in 100 μL of medium/well in white-walled opaque 96-well plates and incubated overnight (37 °C, 5% CO_2_). Conofolidine or the medium alone was added for an additional 24 h. Caspase-Glo 3/7 reagents were then added to each well. After 30 min, relative luminescence was measured in a plate reader (PerkinElmer Life and Analytical Sciences, Buckinghamshire, UK).

### 4.8. Senescence β-Galactosidase Cell Staining Assay

Senescence was investigated using β-Gal staining (Cell Signaling Technology; CST, London, UK). Cells were seeded at a density of 2.5 × 10^4^ cell/well in 6-well plates, incubated overnight at 37 °C, 5% CO_2_, and treated with conofolidine the following day. After 72 h, cells were washed with PBS and fixed before adding 1 mL β-Gal solution (1 mg/mL X-gal, pH = 6.0). Plates were incubated at 37 °C ≥16 h, before being inspected under a light inverted microscope for blue β-Gal staining; positive cells were counted in all wells.

### 4.9. Western Blotting

Following designated treatments, cells were collected and lysed using an NP-40 lysis buffer containing protease and phosphatase inhibitors. Protein concentrations were measured using the Bradford assay [65]. Samples were then loaded (50 μg protein each), and proteins were separated using sodium dodecyl sulphate polyacrylamide gel electrophoresis (SDS-PAGE). By immunoblotting, proteins were transferred onto nitrocellulose membrane, probed against a primary antibody (dilution 1:1000), and thereafter against a secondary antibody (dilution 1:4000). Protein bands were detected using enhanced chemiluminescence (Amersham ECL Prime (Buckinghamshire, UK) and LI-COR Biosciences Ltd. C-DiGit^®^ Blot Scanner, Cambridgeshire, UK). Densitometric analyses of Western blots were performed using Image Studio. Primary antibodies were procured from CST (London, UK): whole and cleaved Poly (ADP-ribose) polymerase (PARP), CDK2, cyclin A2, cyclin D1, p21, NAD(P)H quinone dehydrogenase 1 (NQO1), and β-Actin. A secondary anti-mouse IgG/HRP antibody was purchased from Dako (Cambridgeshire, UK).

### 4.10. Confocal Microscopy

Confocal microscopy was used to visualise the DNA-DSBs foci induced by conofolidine. Cells were seeded in eight-well μ-slides (Ibidi^®^, Gräfelfing, Germany) and allowed to attach overnight in a CO_2_ incubator at 37 °C. Then, they were incubated with either the medium alone or conofolidine or positive control–etoposide. Following a 72 h exposure, cells were fixed with 3.7% formaldehyde in PBS, permeabilised with PBT (0.1% Triton-X-100 in PBS), and incubated with p-Histone γ-H2AX antibody (Sigma-Aldrich, Dorset, UK) for DNA-DSBs foci staining and draq5 (Abcam, Cambridge, UK) for DNA staining. Images were captured using a Zeiss LSM 510 META confocal microscope (Cambridgeshire, UK) and Zeiss LSM 510 image browser (version 4.2). The above protocol was also used for the visual validation of mitotic abnormalities induced by conofolidine except that the β-tubulin antibody TUBB1 (ThermoFisher-Scientific, Oxford, UK) and vincristine—positive control—were used.

### 4.11. Monitoring of γ-H2AX Phosphorylation Induction by Flow Cytometry

Cells were seeded at a density of 1 × 10^6^ in 100 mm^2^ culture Petri dishes, incubated overnight, and treated the following day with either conofolidine 2× GI_50_ or etoposide 2.0 μM—positive control. After 72 h, cells were transferred into FACS tubes containing their respective medium. Cells were centrifuged (10 min; 1500 rpm; 25 °C) and pelleted; supernatants were discarded, and pellets were resuspended again. Thereafter, cells were fixed with 1% methanol-free formaldehyde in PBS, permeabilised with Triton-X-100 in PBS, and incubated with a p-Histone γ-H2AX antibody (Sigma-Aldrich, UK); dilution 1:3333 in 1% FBS in PBS; 1.5 h; 25 °C. For concurrent cell cycle analyses, cells were resuspended in PI (50 μg/mL) and RNase (100 μg/mL) in PBS solution (10 min; 25 °C). The % of γ-H2AX (+) cells and cell cycle distribution were analysed using CyFlow^®^ Space flow cytometer (Sysmex Partec, Goerlitz, Germany) and Kaluza analysis software (Version 2.1, Beckman Coulter, Wycombe, UK).

### 4.12. Reactive Oxygen Species Determination

To measure ROS production, the ROS-Glo H_2_O_2_ (Promega) assay was conducted according to the manufacturer’s protocol, which measures the most stable ROS-H_2_O_2_ level. Cells were plated at a medium density of 5 × 10^3^/70 μL in white-walled, opaque-bottom 96-well plates and incubated overnight (37 °C, 5% CO_2_). Conofolidine or the medium (10 μL) was added to treated or control wells, respectively, for extra 24 h. After 18 h of treatment, 20 μL H_2_O_2_ substrate was added to each well. Upon treatment period completion (24 h), 100 μL ROS-Glo solution was added to each well, and after 20 min, relative luminescence was measured in a plate reader (PerkinElmer Life and Analytical Sciences, Buckinghamshire, UK).

### 4.13. Statistical Analysis

Experiments were repeated at least three times; internal repeats n ≥ 4 (unless otherwise stated) and representative experiments are shown. Data are presented as mean ± SD (or SEM). Statistical differences between study groups were analysed using two-way ANOVA (unless otherwise stated). Dunnett’s multiple comparisons were used to test the significance, determined as *p*-value ≤ 0.05.

## 5. Conclusions

In conclusion, broad-spectrum, potent antitumour activity by bisindole alkaloid conofolidine isolated from *Tabernaemontana corymbosa* has been described. Cancer cell growth and colony formation were significantly suppressed, triggering apoptosis or senescence. However, caution is advised concerning selectivity as activity against non-tumour proliferative cells may result in adverse toxicity. Examination of activity against additional non-cancer cells, tumour spheroids, and co-culture organoids are planned to expand knowledge in this arena. However, such information can only serve as a guide to tumour selectivity, and robust in vivo pharmacokinetic and tolerability/toxicological studies must be performed before any experimental antitumour agent can proceed to in vivo efficacy studies. Cancer selectivity may be enhanced by encapsulation and delivery of conofolidine in nanocarriers, for example, liposomes, or protein nanocages such as apoferritin. Indeed, liposomal formulations of natural products doxorubicin and vincristine are used clinically and have been shown to reduce adverse, systemic, and cardiac toxicities associated with these natural-product “cytotoxic” anticancer agents. The encapsulation and delivery of natural product via apoferritin protein nanocarriers would potentially enhance activity and tumour selectivity through passive (enhanced permeation and retention) and active (transferrin receptor-1) targeting. Indeed, enhanced tumour cell targeting and activity of structurally-related natural product jerantinine A has been achieved following apoferritin encapsulation [66]. Thus, further preclinical evaluation of conofolidine, a promising anticancer candidate, is justified.

## Figures and Tables

**Figure 1 molecules-29-02654-f001:**
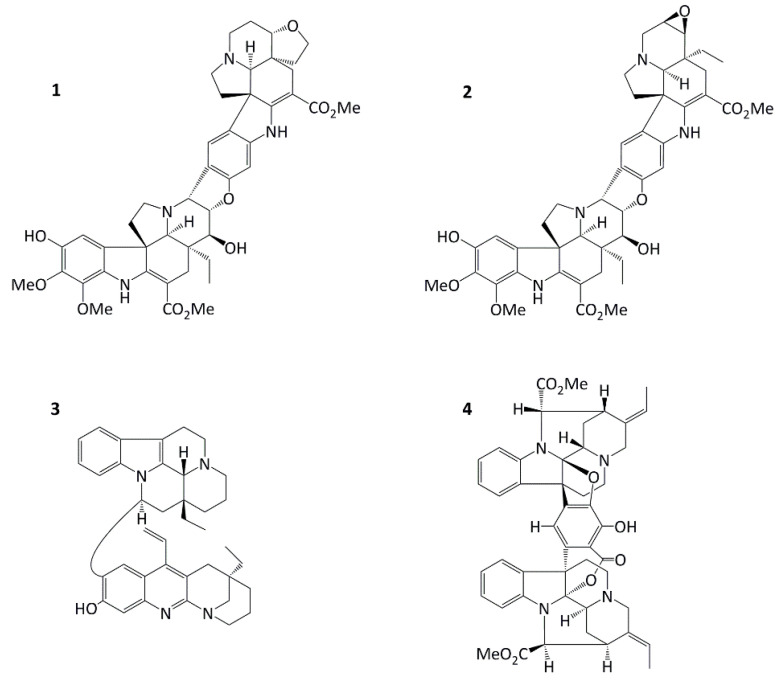
Structures of conofolidine (**1**), conophylline (**2**), leucophyllidine (**3**), and bipleiophylline (**4**).

**Figure 2 molecules-29-02654-f002:**
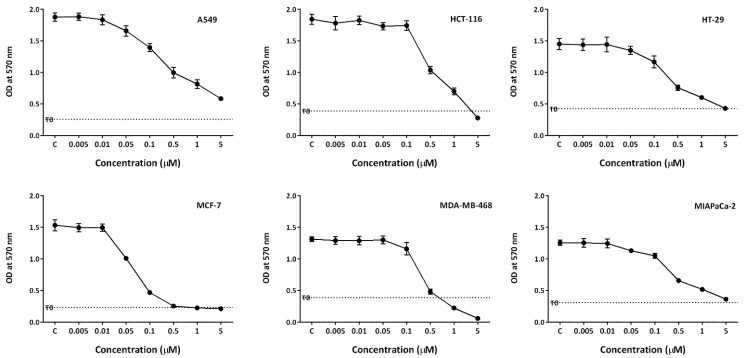
Growth inhibitory effects of conofolidine in A549, HCT-116, HT-29, MCF-7, MDA-MB-468, and MIAPaCa-2. Each representative graph shows one independent MTT trial. Cells were seeded in 96-well plates (3 × 10^3^ cells/well) and treated with TQ for 72 h. No. of trials ≥ 3; n = 4 per independent experiment.

**Figure 3 molecules-29-02654-f003:**
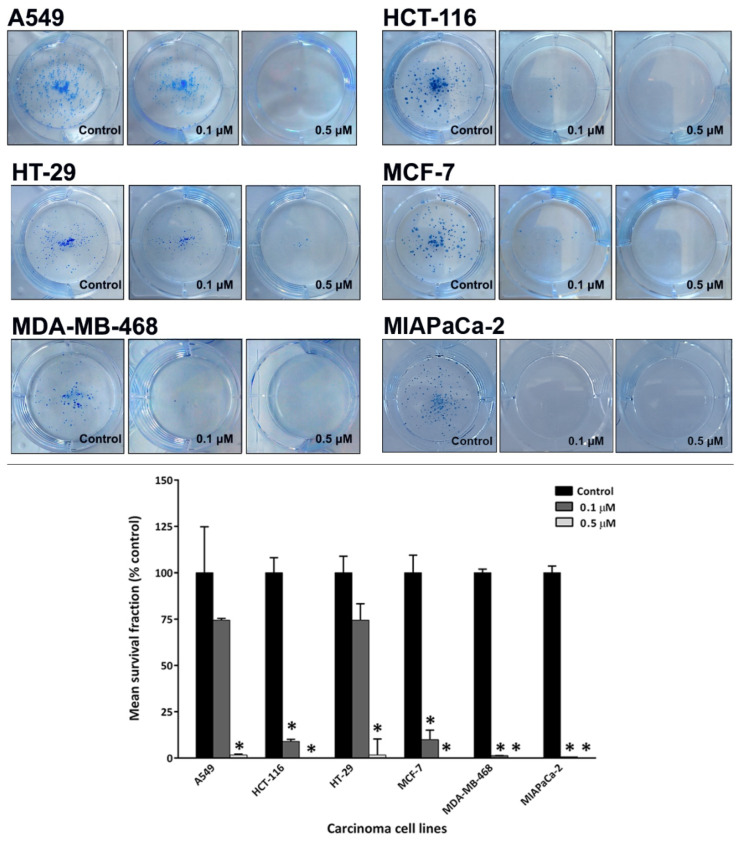
Representative photos of conofolidine’s inhibition of MIAPaCa-2 colonies at 0.1 and 0.5 μM. Mean ± SD bars show mean survival fraction as a % of the control. Asterisk indicates significant inhibition (*p* ≤ 0.05). Cells were seeded and treated with conofolidine (24 h), and then their medias were replaced with fresh ones. When colonies contained ≥50 cells, the colonies were fixed, stained, and counted. Plating efficiencies ranged between ~25% and ~65%. No. of trials ≥ 3; n = 2 per independent experiment.

**Figure 4 molecules-29-02654-f004:**
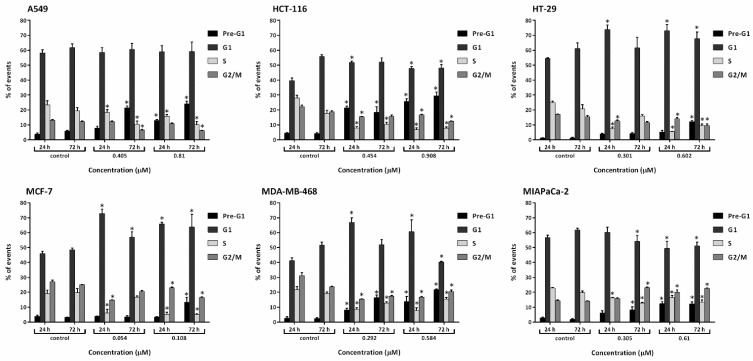
Mean ± SD bars showing % distribution of cell cycle phases of A549, HCT-116, HT-29, MCF-7, MDA-MB-468, and MIAPaCa-2 following 24, 72 h-1×, and 2× GI_50_ conofolidine treatments. S-phase depletion and pre-G1 increase are the most significantly encountered changes. Cells were treated and then stained with PI, and at least 20,000 events/sample were measured. Asterisks indicate significant (*p* ≤ 0.05) changes compared to the control. No. of trials ≥ 3; n = 2 per independent experiment.

**Figure 5 molecules-29-02654-f005:**
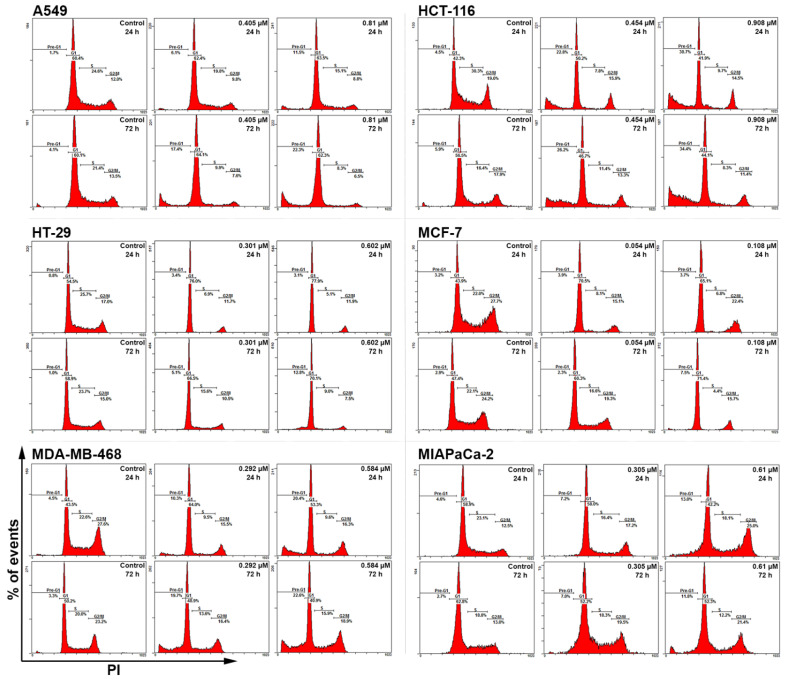
Representative cell cycle histograms showing conofolidine-induced perturbations in A549, HCT-116, HT-29 MCF-7, MDA-MB-468, and MIAPaCa-2 cell cycles after 24 and 72 h-treatments with their respective 1× and 2× GI_50_. The most prominent observations are depleted S-phase with occasional G1-arrest and increased pre-G1(<2 N) events. The G1/S arrest is mainly observed in HT-29 and MCF-7 treatment groups. An exceptionally increased G2/M can also be seen in MIAPaCa-2 cells. Cells were treated and then stained with PI, and at least 20,000 events/sample were measured. No. of trials ≥ 3; n = 2 per independent experiment.

**Figure 6 molecules-29-02654-f006:**
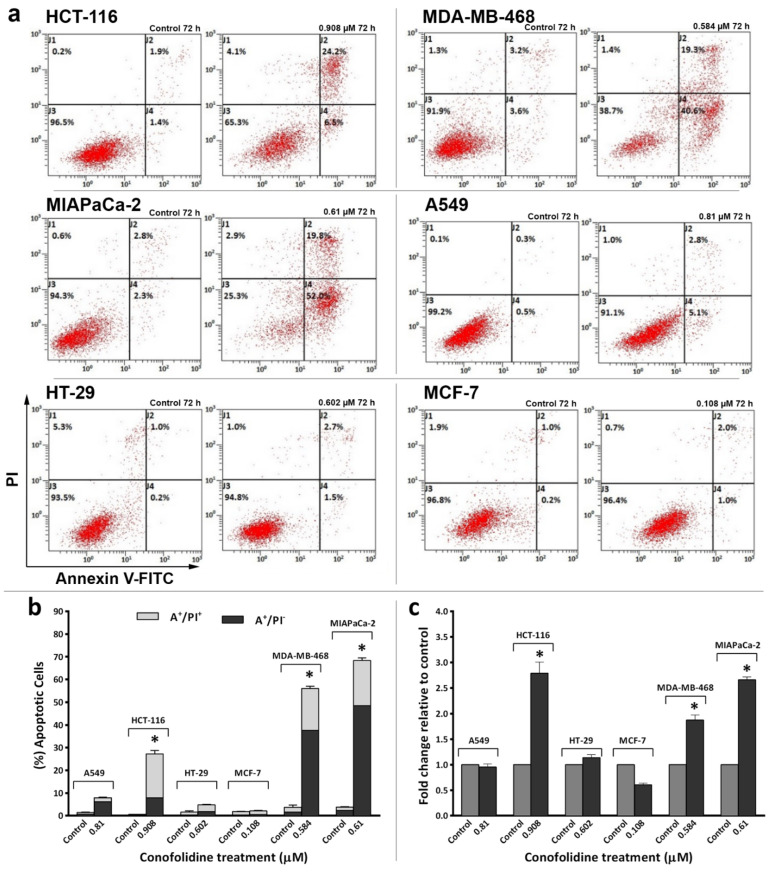
(**a**,**b**) Representative dot-plots and mean ± SD bars showing annexin V/PI results of carcinoma cells treated with conofolidine for 72 h with respective 2× GI_50_. Significantly increased apoptotic cells compared to the control could only be seen in MDA-MB-468, MIAPaCa-2, and HCT-116 but not in HT-29. Like HT-29, A549 and MCF-7 did not show significant apoptotic events. Samples were stained with annexin V/PI, and at least 10,000 events were detected. The percentage of apoptotic events was equal to the sum of cells undergoing early-apoptosis (AV+/PI−) plus late-apoptosis (AV+/PI+). Asterisk indicates statistically significant (*p* ≤ 0.05) increments compared to the control. No. of trials ≥ 3; n = 2 per independent experiment. (**c**) Mean ± SD bars showing changes in caspases 3/7 activities following 24 h conofolidine (2× GI_50_) exposure in carcinoma cell lines. Conofolidine significantly increased caspases 3/7 activities (shown as fold change relative to the control) in HCT-116, MDA-MB-468, and MIAPaCa-2. However, no significant changes were seen in A549, HT-29, and decreased activities were observed in MCF-7 cells. Assays were repeated 3 times (n = 2). Asterisks indicate a statistically significant (*p* ≤ 0.05) change compared to the control.

**Figure 7 molecules-29-02654-f007:**
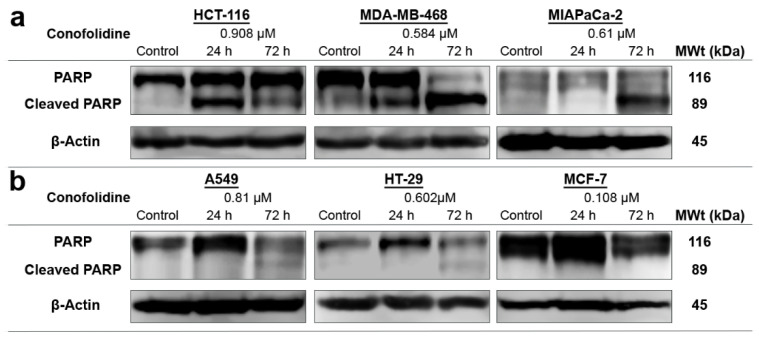
Representative Western blot bands showing expression of whole and cleaved PARPs in protein lysates of carcinoma cell lines treated with conofolidine (2× GI_50_, 24 h and 72 h). Western blots were performed using antibodies to detect whole and cleaved PARPs and housekeeping gene β-Actin. (**a**) Cleaved PARP can be seen in HCT-116 after 24 h and in MDA-MB-468 and MIAPaCa-2 cells after 72 h exposures. (**b**) Faint bands of cleaved PARP can be seen in A549 and HT-29 cells after 72 h while no cleaved PARP can be seen in MCF-7. Assays were repeated 3 times.

**Figure 8 molecules-29-02654-f008:**
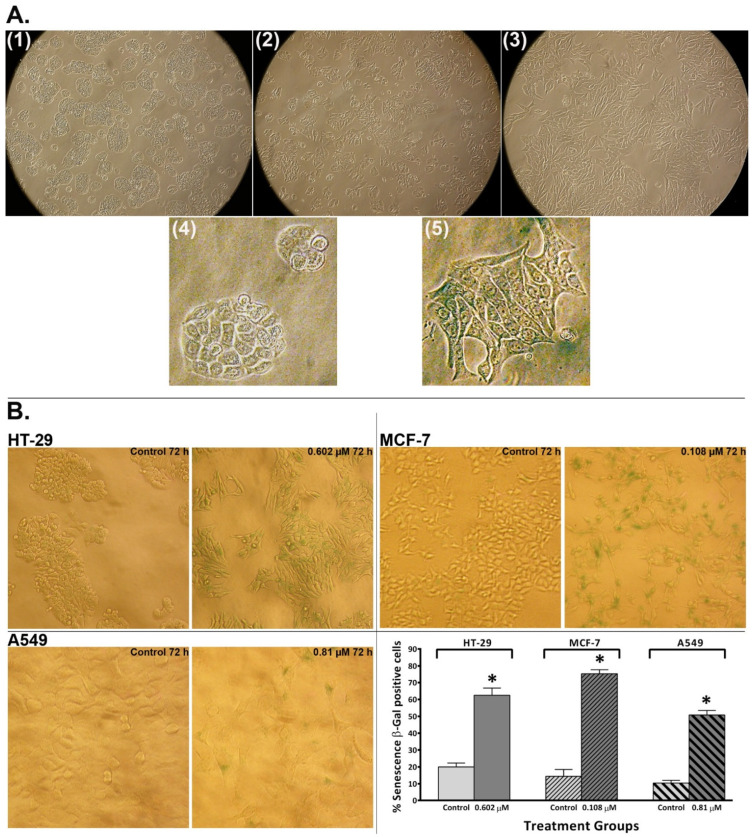
(**A**) Representative images of HT-29 cells from a light inverted microscope showing morphological and size alterations induced by conofolidine (2× GI_50_). (**1**) ~65% confluent untreated control cells, (**2**) 24 h conofolidine-treated cells, and (**3**) 72 h conofolidine-treated cells at 10× objective magnification. Examples of cells’ morphology following 72 h incubation with the vehicle control (**4**) and 2× GI_50_ conofolidine (**5**); 40× objective magnification. (**B**) Representative images from a light inverted microscope and mean ± SD bars showing increased % of senescence-associated β-Gal positive cells induced by conofolidine (2× GI_50_, 72 h) compared to untreated controls HT-29, MCF-7, and A549. Images of both HT-29 and MCF-7 cells were captured at 20× while A549 images were taken at 40× objective magnifications. Mean bars ± SD showing significant (*p* ≤ 0.05) increase in the percentage of senescence-associated β-Gal positive cells relative to untreated controls in HT-29, MCF-7, and A549 cells treated with conofolidine (2× GI_50_, 72 h). Cells were either incubated for 72 h with a medium alone or with conofolidine (2× GI_50_) and then stained with β-Gal, after which images were captured from at least 4 different areas in each well of treatment groups. From captured images, senescence β-Gal positive cells were counted in both treated and untreated controls, and at least 1000 cells were counted per treatment group. Assays were repeated 3 times (n = 2). Asterisks indicate a statistically significant (*p* ≤ 0.05) change compared to the control.

**Figure 9 molecules-29-02654-f009:**
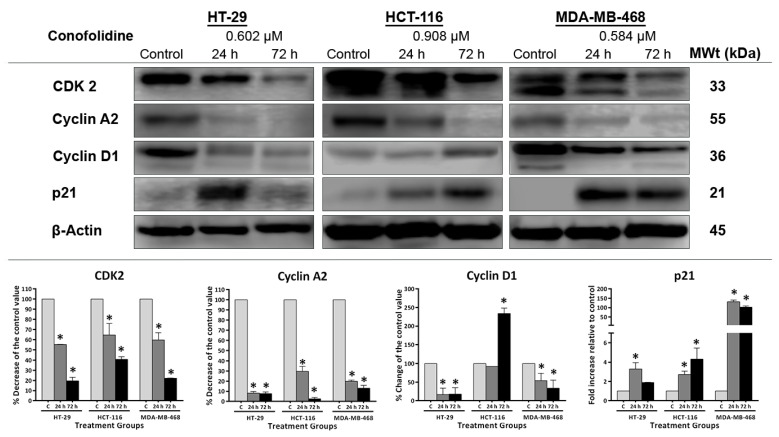
Representative Western blot showing expression of and mean ± SD error bars from densitometric analysis of CDK2, cyclin A2, cyclin D1, and p21 in protein lysates of HT-29, HCT-116, and MDA-MB-468 cells treated with conofolidine (24 h and 72 h, 2× GI_50_). Western blots were performed using antibodies against CDK2, cyclin A2, cyclin D1, p21, and housekeeping gene β-Actin. Mean ± SD error bars are presented as the % decrease of the control value of CDK2, cyclin- A2, and D1 and as the fold increase in p21 relative to the control. Conofolidine significantly decreased CDK2 and cyclin A2 and increased p21 expressions. Cyclin D1 is also decreased in HT-29 and MDA-MB-468 while its expression is increased in HCT-116 cells. No changed β-Actin expression was seen. Assays were repeated 2 times. Asterisks indicate a statistically significant (*p* ≤ 0.05) change compared to the control.

**Figure 10 molecules-29-02654-f010:**
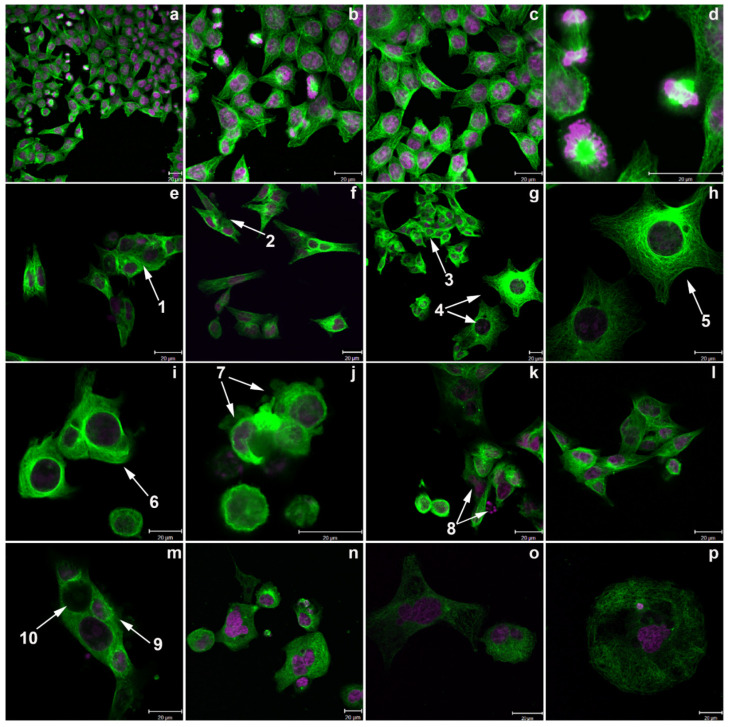
Confocal images showing HCT-116 cells treated with conofolidine compared to the untreated control and vincristine-treated cells. (**a**–**d**) Controls with a medium only; (**e**–**k**) conofolidine 1× GI_50_-treated cells; (**l**,**m**) conofolidine 2× GI_50_-treated cells; (**n**–**p**) vincristine (10 ηM)-treated cells. Conofolidine induced variable morphological phenotypes including chaotic multiplication with occasional multi-nucleation (arrows **1**–**3**), multipolar large cells (arrows **4**,**5**), loss of polarity with asymmetric division (arrow **6**), membrane blebbing and/or DNA-fragmentation (arrows **7**–**9**), and cells with varying DNA contents (arrow **10**). Vincristine-treated HCT-116 cells showed reduced microtubule density, microtubule irregularity with multipolar spindles, and abundant multi-nucleation.

**Figure 11 molecules-29-02654-f011:**
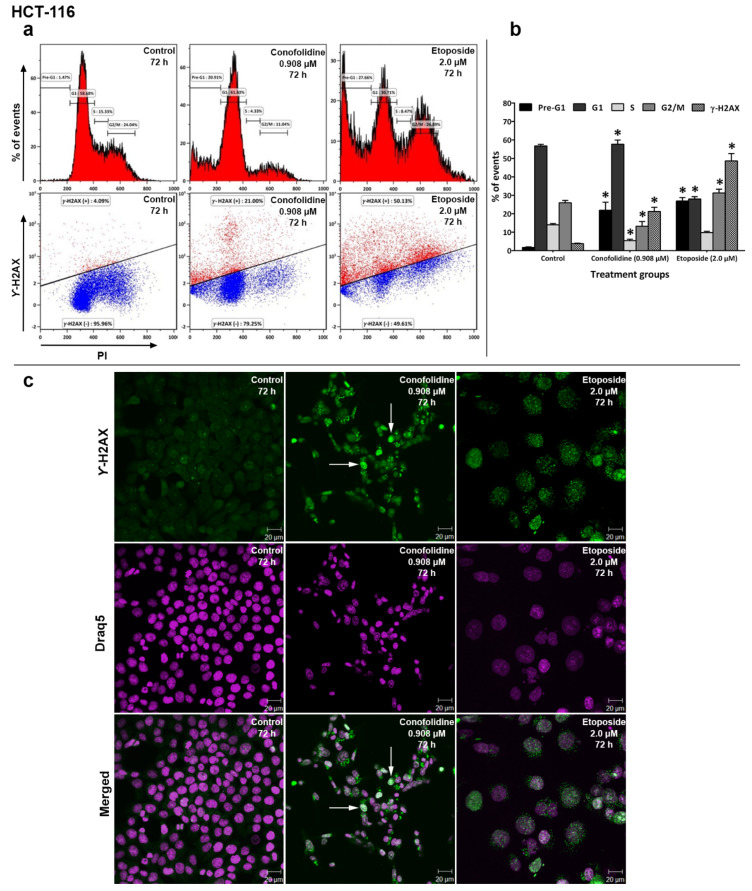
(**a**) Representative histograms and dot-plots from independent trial showing HCT-116 cell cycle perturbation and γ-H2AX induction after 72 h, 2× GI_50_ conofolidine treatment. The majority of DSBs were seen in the G1 phase of the cell cycle compared to controls and the etoposide (positive control)-treated cells. (**b**) Mean ± SD bars showing changes in the percentage of the HCT-116 cell cycle phases and increased DNA-DSBs following 72 h, 2× GI_50_ conofolidine exposure compared to the untreated control. Asterisks indicate a statistically significant (*p* ≤ 0.05) change compared to the control. No. of trials ≥ 3; n = 2 per independent experiment. At least 15,000 events were obtained/sampled. (**c**) Confocal images from one trial of HCT-116 treated with conofolidine showing γ-H2AX foci (green fluorescence) compared to the control. Etoposide-treated cells also showed γ-H2AX foci formation. Arrows show examples of nuclei containing γ-H2AX foci saturation. A smaller DNA content was seen in each of conofolidine-treated cells compared to the control. Assays were repeated 2 times, and at least three images were obtained from each treatment.

**Figure 12 molecules-29-02654-f012:**
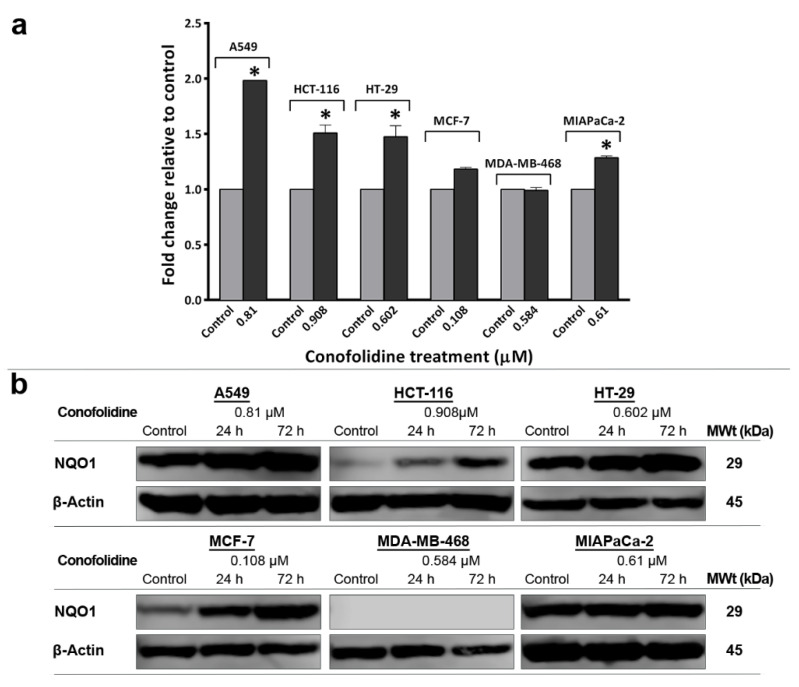
(**a**) Mean ± SD error bars showing fold change relative to the control of ROS level in carcinoma cells following conofolidine treatment (2× GI_50,_ 24 h). Significantly increased ROS is seen in A549, HCT 116, HT-29, and MIAPaCa-2; MCF-7 shows a slight increment while no significant change is observed in MDA-MB-468. Assays were repeated 3 times (n = 2). Asterisks indicate a statistically significant (*p* ≤ 0.05) change compared to the control. (**b**) Representative Western blot showing expression of NQO1 in protein lysates of A549, HCT 116, HT-29, MCF-7, MDA-MB-468, and MIAPaCa-2 cells treated with conofolidine (24 h and 72 h, 2× GI_50_). Western blotting was performed using antibodies against NQO1 and housekeeping gene β-Actin. All cell lines (except MDA-MB-468) showed increased NQO1 expression following conofolidine treatment. No changed β-Actin expression was seen. Assays were repeated 2 times.

**Table 1 molecules-29-02654-t001:** Bisindoles’ anti-proliferative effects on human carcinoma cell lines.

Human Cell Line		72 h Exposure MTT GI_50_ (μM)			
Origin	Designation	Conofolidine	Conophylline	Leucophyllidine	Bipleiophylline
Breast carcinoma	MCF-7	0.054 ± 0.08	0.066 ± 0.03	2.79 ± 0.98	3.49 ± 0.47
	MDA-MB-468	0.292 ± 0.03	0.557 ± 0.08	1.99 ± 0.03	5.81 ± 1.11
Colon carcinoma	HCT-116	0.454 ± 0.07	0.737 ± 0.27	2.43 ± 0.08	3.86 ± 1.04
	HT-29	0.301 ± 0.03	0.350 ± 0.52	2.48 ± 0.17	7.72 ± 4.08
Pancreatic carcinoma	MIAPaCa-2	0.305 ± 0.03	0.663 ± 0.03	3.29 ± 0.98	12.37 ± 3.12
Lung carcinoma	A549	0.405 ± 0.07	1.743 ± 0.34	4.45 ± 0.27	13.63 ± 1.94

GI_50_ values were estimated from MTT data after 72 h of exposure to bisindole alkaloids (n = 4) and expressed as mean ± SD of three independent trials.

## Data Availability

The data presented in this study are available in the article and Appendix A.

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
