# Peer review of "Conofolidine: A Natural Plant Alkaloid That Causes Apoptosis and Senescence in Cancer Cells"

_molecules, 2024, doi:10.3390/molecules29112654_

Round 1
Reviewer 1 Report
Comments and Suggestions for Authors
Dear Authors, your work highlights for the first time the anticancer role of the bisindole alkaloid conofolidine. Many of the observations reported are interesting even if you have not sufficiently discussed that this compound shows a powerful inhibitory effect on the growth also of MCF10A, making its future potential application in breast cancer certainly less attractive. Furthermore, the manuscript requires some modifications to align with the journal's suggestion for publication. See the major comment below. For this reasons, I believe that the manuscript requires major revisions before publication.
Major comments:
1) In my opinion, simple summery should be delated because it repeats what is described in the abstract.
2) The materials and methods section should be moved after the discussion section, as indicated in the journal's instruction fot authors.
3) Lines 94-96 A curiosity of mine: Why did you decide to use RPMI medium for all cell lines? Some of these cell lines are usually grown in DMEM.
4) Unfortunately, the growth inhibitory effect of conofolodine is also very powerful on the benign epithelial breast cells MCF10 A. This aspect is not discuss in the manuscript but in my opinion it is very important.
5) Figure S6 should be incorporated into Figure 3.
6) I don’t understand the need to create Figure S7 when the flow cytometry results of A549 and MCF-7 should be presented in Figure 6 along with the data on the other cell lines. Furthermore, in the text of the legend of figure S7 among the cell lines evaluated also appear the name of HT-29 cells which actually are present in figure 6.
7) In my opinion, the result 3.6 should be introduced differently. Why do you start by talking about the morphology of HT-29 cells only? Perhaps the hypothesis of the induction of senescence should be introduced differently, talking about all the cell lines in which you don’t see apoptosis induction. I wouldn't even talk about differentiation induction since you only evaluated it on the HT-29 and for this reason I would remove figure S8. Figures 8 and 9 could be brought together in a single figure with photos of the morphology of all cell line (HT29, MCF-7 and A549) and results of beta-gal.
8) In figure 11 I wouldn’t talk about abundant multi-nucleation after conofolidine treatment (e-g). Abundant multi-nucleation is very clear only in vincristine-treated cells (n-p).
9) The file containing original images in not correct because it contains the same cropped images of Western blot presented in figure 7, 10 and 13. In this file you have to upload all original uncropped images corresponding the Western blot presented in the manuscript.
Minor comments:
1) Line 69 - Change the numbers in brackets next to Figure 1 (1-4 instead of 10-13).
2) Line 110 – MTT dose missing.
3) In the legends of figures S1 and S2 it is written that cells were treated with conofolidine instead of conophilline, leucophyllidine and bipleiophylline respectively.
4) Figure S4 could be removed as it says nothing more than Table 1 and Figures 1, S1 and S2.
Author Response
Dear Editorial Board “Molecules”,
Authors are very grateful for comments and revisions suggested for manuscript “Conofolidine, a natural plant alkaloid causes apoptosis and senescence in cancer cells.”.(ID: molecules-2831252).
We have addressed reviewers` comments, for which authors are grateful, and hope that the referees and editorial board agree that a robust revision has been generated as a result of reviewers` thoughtful suggestions.
Please see itemised below changes made in response to reviewers` comments.
Reviewer 1.
- In my opinion, simple summary should be deleted because it repeats what is described in the abstract.
The simple summary has been deleted.
- The materials and methods section should be moved after the discussion section, as indicated in the journal`s instruction for authors.
Materials and Methods section has been repositioned to follow the discussion.
- Lines 94-96 a curiosity of mine: Why did you decide to use RPMI medium for all cell lines? Some of these cells lines are usually grown in DMEM.
Thank you for raising this interesting point. We were following ATCC instructions and continuing protocols established in our laboratory and successfully adopted. We note that cell lines grow equally well in both media; indeed some current students – following current guidelines have begun to grow certain cell lines in DMEM. However, for this work, we sought to maintain consistency in the growth conditions for cell lines used to avoid as far as possible additional variations that could be introduced by the use of different media. Thus, we continued to use a culture medium that is suitable for the majority of cell lines included in our work. RPMI1640 meets this criterion as its commonly used for most types of adherent and suspension mammalian cells, including tumour cells, primary culture cells, and non-transformed cells. Additionally, there are disadvantages associated with DMEM relating to the content of Phenol red which possesses oestrogenic properties and high fluorescence, factors which may impact cell proliferation (in cells that express oestrogen receptors) and background fluorescence readings respectively.
- Unfortunately, the growth inhibitory effect of conofolidine is also very powerful on the benign breast cells MCF10A. This aspect is not discussed in the manuscript and in my opinion it is very important.
Thank you for pointing this out. Yes, authors agree this is a very important point and we apologise for the omission. There are 2 pertinent discussion points that have been included. MCF-10A is a cell line derived from fibrocystic disease; it is non-tumourigenic, but highly proliferative, so not `normal`. This fact is discussed (lines 371-372, discussion). In the closing conclusion (lines 688-703), we consider the possibility of enhancing cancer-selectivity by encapsulation and delivery of conofolidine in nanocarriers for example liposomes or protein nanocages such as apoferritin. Indeed liposomal formulations of natural products doxorubicin and vincristine are used clinically and have been shown to reduce adverse, systemic as well as cardiac toxicities associated with these natural product `cytotoxic` anticancer agents. Encapsulation and delivery of `cytotoxic` natural product via apoferritin protein nanocarrier would potentially enhance activity and tumour selectivity through passive (EPR) and active (transferrin receptor-1) targeting, accumulating preferably in tumour cells, allowing administration of lower doses. In our own labs, we have shown enhanced tumour cell targeting and activity of natural product jerantinine A following apoferritin encapsulation. Reference Abuzaid et al, 2022 has been added in the final conclusion. Another strategy to enhance tumour targeting is through antibody drug conjugation, as has been achieved with Kadcyla following conjugation of the natural product emtansine to Herceptin.
- Figure S6 should be incorporated into Figure 3.
Authors thank reviewer 1 for pointing this out. We have incorporated Figure 6 into Figure 3.
- I don`t understand the need to create Figure S7 when the flow cytometry results of A549 and MCF-7 should be presented in Figure 6 along with the data on the other cell lines. Furthermore, in the text of the legend of Figure S7 among the cell lines evaluated also appear the name of HT-29 cells which actually are present in Figure 6.
Thank you for pointing this out, we agree and have incorporated Figure S7 into Figure 6.
- In my opinion, the result 3.6 should be introduced differently. Why do you start by talking about the morphology of HT-29 cells only? Perhaps the hypothesis of the induction of senescence should be introduced differently, talking about all the cell lines in which you don`t see apoptosis-induction. I wouldn`t even talk about differentiation-induction since you only evaluated it in HT-29 and for this reason I would remove Figure S8. Figures 8 and 9 could be brought togetherin a single figure with photos of the morphology of all cell lines (HT-29, MCF-7 and A549) and results of beta-gal.
Thank you for your thoughtful consideration of this point. Indeed, we hypothesized differentiation-induction because we know that conophylline (conofolidine group prototype) is a known differentiation inducer in pancreatic beta cells, thus we initially thought that the clear morphological changes brought about by conofolidine in HT-29, consistent with reported morphological changes associated with differentiation, may be a consequence of differentiation-induction in this colorectal carcinoma cell line. We consequently tested (and rejected) the hypothesis that differentiation was being induced, before investigating senescence-induction.
With respect to merging figures 8 & 9, thank you for this suggestion, accordingly we have brought them together in a single figure (figure 8), and have amended the subsequent figure sequence (figure 10 becomes figure 9; figure 11 becomes figure 10 and so on).
- In Figure 11, I wouldn`t talk about abundant multi-nucleation after conofolidine treatment. Abundant multi-nucleation is very clear only in vincristine-treated cells.
Thank you for pointing this out. Accordingly, we have changes `abundant` to `occasional` multi-nucleation after conofolidine treatment, and added `abundant` multi-nucleation to the explanation of vincristine-treated cells only.
- The file containing original images is not correct because it contains the same cropped images of Western blot presented in Figures 7, 10 and 13. In this file you have to upload all original uncropped images corresponding to the Western blot presented in the manuscript.
Authors are sincerely sorry, but sadly, we no longer have the whole blots. The western blots were carried out by Mohammed Al-Hayali when he was a PhD student at the University of Nottingham within the School of Pharmacy between the years of 2014 – 2018. The figure presented are images of the original blots. Mohammed no longer has is laptop on which the images were stored. You are very welcome to have a copy of Mohammed`s thesis in which are copies of the original blots; data were checked and verified by internal and external assessors.
Minor comments:
- Line 69 – Change the numbers in brackets next to Figure 1 (Figure 1-4 instead of 10-13).
Apologies, this has now been changed.
- Line 110 – MTT dose is missing.
Thank you for pointing this out, we have accordingly changed it and as the materials and methods are moved after discussion it becomes line 589.
- In the legends of Figures S1 and S2 it is written that cells were treated with conofolidine instead of conophylline, leucophyllidine and bipleiophylline respectively.
Thank you for pointing out this error, we have accordingly corrected the legends of Figures S1 and S2.
- Figure S4 can be removed as it says nothing more than Table 1 and Figures 1, S1 and S2.
Thank you, respectfully, do you mean Figure S3? This (Fig S3) has now been removed, and subsequent figure numbers altered accordingly.
Reviewer 2
This manuscript reports conofolidine`s anticancer activity against human-derived carcinoma cell lines. The abstract is not clear, for example cell lines used fofr the tests are not reported. The introduction must be improved. Analysis of recent literature on natural products as new anticancer drugs should be implemented. There are many examples of anticancer molecules extracted from plants and their bioactivity could be important to compare it with conofolidine`s anticancer activity and selectivity.
Authors thank reviewers 2 for these comments and have made clear in the abstract specific cell line origins and names.
More current literature has been introduced into the manuscript e.g. the introduction, line 42 includes Wang et al Sokotrasterol; review Talib et al, and Al-Hayali et al work on thymoquinone isolated from Nigella sativa. These become references 3,4,5. Additionally, Chaib et al 2022 has been added to the introduction, together with brief rational introduction to senescence-induction and senolysis as a therapeutic strategy. (lines 51-55 introduction). In the final conclusion, Abuzaid et al (2021) has been cited, a recent publication describing apoferritin-encapsulation of natural product jerantinine to enhance cancer selectivity.
In the introduction the aim of the research work reported in the paper must be better described.
The aims of the research are stated in lines 58-62, prior to brief summary of research findings.
The procedure used for the production of the extracts is not reported. Authors should describe it or add a reference describing the procedure. It is also necessary for authors to indicate details regarding the collection of Tabernaemontana corymbose Roxb leaves (time of year, cultivation, place of origin etc), because the type and quantity of bioactive products extracted depends on these factors. In fact, authors affirm that conofolidine increased oxidative stress, preceding apoptosis- and senescence-induction in most carcinoma cell lines tested by enhanced ROS levels accompanied by NQO1 expression, but they have not studied the effect of this molecule towards normal cell. Therefore, before considering this molecule as a possible anti-cancer drug, its bioactivity should be tested on primary cells to exclude that the same effects (induction of apoptosis and senescence) can also be induced on healthy cells.
Nge et al 2016 (Professor Toh-Seok Kam`s group) report isolation and identification of conofolidine in detail. This reference is included in the manuscript and bibliography – originally reference 14 (revised reference 18). Authors agree with reviewers 2, that the bioactivity of conofolidine should be thoroughly investigated in normal tissues and cells. The growth inhibitory effects of conofolidine have been reported here in two non-cancer cell lines – MRC-5 fibroblasts and MCF-10A fibrocystic proliferative breast cells, as an initial exploration into the cancer-selectivity of conofolidine. Effects on `normal` cells and tissues are an important goal of future preclinical evaluation of conofolidine. Strategies to mitigate adverse effects on non-cancer cells are considered in the final conclusion (lines 699-708).
In all the reported Figures must be specified positive and negative control.
Thank you. Vehicle alone served as negative controls. Positive controls included vincristine (Figure 10, for confocal images). It is known that the microtubule disrupting agent vincristine will cause microtubule disruption, evoke multipolar spindles and allow detection of apoptotic characteristics (membrane blebbing), allowing comparison. Etoposide was also used as a positive control as a known apoptosis-inducer and as a natural product that causes DNA double strand breaks. In assays to detect g-H2AX, and dual annexin V/PI (apoptosis assays) etoposide was selected as a positive control.
In addition, the positive control sodium butyrate was used as a known differentiation-inducer (Supplementary information Figure S5).
More references must be reported.
Please refer to earlier comments. Briefly the following new references have been included: Wang et al, 2024; Al Hayali et al, 2021; Talib et al, 2020; Chaib et al, 2022; Abuzaid et al, 2021.
There are many inaccuracies in the text that must be carefully read and corrected. Some inaccuracies are reported below.
Thank you for careful attention to detail and pointing out these inaccuracies, which have now been corrected.
Line 34: please report the name of cell lines.
The names of cell lines have been added as follows:
“The DNA integrity assessment of HCT-116, MDA-MB-468, MIAPaCa-2 and HT-29…”
Add a paragraph named “Materials” describing materials used for the experiments and the supplier (Merck? Millipore? Etc).
Thank you for your comments. According to the guidance, Materials and Methods section should be together, not separated? We have included suppliers` details (e.g. MTT, Alfa-Aesar, Lancashire, UK).
Lines 88-89: please describe how the extracts are obtained and when and where the Tabernaemontana corymbose Roxb leaves were collected.
We have included in the methods section the reference detailing collection, extraction, isolation and purification – Nge et al, 2016.
Lines 96-97: please report the name of each cell line of the kind of cells (ie breast cell cancer, colon cell cancer etc).
Thank you for your comment. Done accordingly as follows:
“Cell lines included are breast carcinoma - MCF-7, MDA-MB-231, MDA-MB-468, SK-BR3, T-47D and ZR-75-B; colorectal carcinoma - Caco-2, HCT-116 and HT-29; lung carcinoma - A549; pancreatic carcinoma MIAPaCa-2 and PANC-1; lymphoma - DoHH2 and Vallois.”
Please note that materials and methods section has been moved to the end of the paper and becomes 4.1, 4.2,…etc; starting from line 559.
Line 128: please remove 1991 as you report the reference number.
1991 has been removed.
Line 129: (1.0 × 105 and 0.5 × 105 cells/well for 24 h and 72 h exposures, respectively): the right exponential notation.
Thank you, the correct notation has been adopted throughout the methods section.
Lines 132, 141: 1200 rpm, not 1,200 rpm
Corrected.
Line 138: 0.5-0.75 × 105. This is not correct, the right exponential notation and the right numbers are 5.0-7.5 x 106
0.5-0.75 × 105 has been changed to 5-7x104
Line 160: (London, UK)). Please remove the second bracket.
The second bracket has been removed.
Many thanks for your consideration of this manuscript.
Sincerely, on behalf of all authors,
Tracey Bradshaw,
Associate Professor,
University of Nottingham.
Reviewer 2 Report
Comments and Suggestions for Authors
This manuscript reports confolidine’s anticancer activity against human-derived carcinoma cell lines. The abstract is not clear, for example cell lines used for the tests are not reported. The introduction must be improved. Analysis of recent literature on natural products as new anticancer drugs should be implemented. There are many examples of anticancer molecules extracted from plants and their bioactivity could be important to compare it with confolidine’s anticancer activity and selectivity.
In the introduction the aim of the research work reported in the paper must be better described.
The procedure used for the production of the extracts is not reported. Authors should describe it or add a reference describing the procedure. It is also necessary for the Authors to indicate details regarding the collection of Tabernaemontana corymbosa Roxb leaves. (time of year, cultivation, place of origin, etc.), because the type and quantity of bioactive products extracted depends on these factors. In fact, Authors affirm that Conofolidine increased oxidative stress, preceding apoptosis- and senescence-induction in most carcinoma cell lines tested by enhanced ROS levels accompanied by NQO1 expression but they have not studied the effect of this molecule towards normal cell. therefore, before considering this molecule as a possible anti-cancer drug, its bioactivity should be tested on primary cells to exclude that the same effects (induction of apoptosis and senescence) can also be induced on healthy cells.
In all the reported Figures must be specified positive and negative control.
More recent Refernces must be reported.
There are many inaccuracies in the text that must be carefully read and corrected. Some inaccuracies are reported below
Line 34: please report the name of cell lines
Add a paragraph named “Materials” describing materials used for the experiments and the supplier (Merck? Millipore? Et.)
Lines 88-89: please describe how the extracts are obtained and when and where the Tabernaemontana corymbosa Roxb leaves were collected
Lines 96-97: please report the name of each cell line or the kind of cells (i.s. breast cell cancer, colon cell cancer, etc.)
Line 128: please remove 1991 as you report the reference number
Line 129: (1.0 × 105 and 0.5 × 105 cells/well for 24 h and 72 h exposures, respectively): the right exponential notation is 5x106
Lines 132, 141: 1200 rpm, not 1,200 rpm
Line 138: 0.5-0.75 × 105. This is not correct, the right exponential notation and the right numbers are 5.0-7.5 x 106
Line 160: (London, UK)). Please remove the second bracket.
Author Response

(The authors gave the same response as above.)

Round 2
Reviewer 1 Report
Comments and Suggestions for Authors
Dear Authors, I really appreciated all the changes you made to the manuscript thus making it clearer and more fluent. However, I remained very surprise and disconcerted by the impossibility of recovering the original uncropped images of the numerous Western blots that you have presented in your paper. I will leave the Editor with a final comment regarding this point
Author Response
Authors thank the reviewers for their comments, for acknowledgement and appreciation of changes made; we are sincerely sorry for the omission of Western blots. We acknowledge that this is not ideal and would dearly like to be able to present the whole of the original blots. However, as explained, the blots shown are cropped sections of the original (unmodified) blots performed by Dr. Al-Hayali during his PhD studies. I would like to assure reviewers that Dr. Al Hayali conducted the most meticulous laboratory experiments and his data produced were to an exceptionally high standard. Unfortunately, since this period, Dr. Al-Hayali has travelled back to his own country and his computer withheld, thus data stored are lost. The systems at the university have been updated and the stored files seem irretrievable. Dr. Al-Hayali received his PhD in 2018. His data were scrutinized, reviewed and verified by internal and external examiners; blots are presented in his PhD thesis, which we are very happy to share with Molecules` reviewers. The thesis is freely available from The University of Nottingham repository.
Reviewer 2 Report
Comments and Suggestions for Authors
The anticancer activity of the tested compounds is well demonstrated, the methods adequately described but the research design should be improved by performing the same experiments also on normal cells in order to study the cytotoxic activity of the samples and verify whether they are selective for tumor cells .
Author Response
Authors agree that experiments on normal and/or additional non-cancer cells should be performed, and indeed design of such experiments are included in plans for further preclinical evaluation – together with in vitro examination of activity in 3D tumour models such as spheroids and co-culture organoids. However, this can only serve as a guide to tumour selectivity, and robust in vivo pharmacokinetic and tolerability/toxicological studies must be performed before an experimental antitumour agent can be proceeded to in vivo efficacy studies. In the present study we report examination of activity in 2 non-cancer models – MRC5 fibroblasts and MCF-10A non-tumorigenic breast cells derived from a patient with fibrocystic proliferative disease. Thank you for your consideration.